# Interaction of Vinyl-Type Carbocations, C_3_H_5_^+^ and C_4_H_7_^+^ with Molecules of Water, Alcohols, and Acetone

**DOI:** 10.3390/molecules28031146

**Published:** 2023-01-23

**Authors:** Evgenii S. Stoyanov, Irina Yu. Bagryanskaya, Irina V. Stoyanova

**Affiliations:** N.N. Vorozhtsov Institute of Organic Chemistry, Siberian Branch of Russian Academy of Sciences, 630090 Novosibirsk, Russia

**Keywords:** vinyl cations, vinyl cation adduct, very strong H-bond, proton disolvate, carborane salt

## Abstract

X-ray diffraction analysis and IR spectroscopy were used to study the products of the interaction of vinyl cations C_3_H_5_^+^ and C_4_H_7_^+^ (Cat^+^) (as salts of carborane anion CHB_11_Cl_11_^−^) with basic molecules of water, alcohols, and acetone that can crystallize from solutions in dichloromethane and C_6_HF_5_. Interaction with water, as content increased, proceeded via three-stages. (1) adduct Cat^+^·OH_2_ forms in which H_2_O binds (through the O atom) to the C=C^+^ bond of the cation with the same strength as seen in the binding to Na in Na(H_2_O)_6_^+^. (2) H^+^ is transferred from cation Cat^+^·OH_2_ to a water molecule forming H_3_O^+^ and alcohol molecules (L) having the CH=CHOH entity. The O- atom of alcohols is attached to the H atom of the C=C^+^-H moiety of Cat^+^ thereby forming a very strong asymmetric H–bond, (C=)C^+^-H⋅⋅⋅O. (3) Finally all vinyl cations are converted into alcohol molecule L and H_3_O^+^ cations, yielding proton disolvates L-H^+^-L with a symmetric very strong H-bond. When an acetone molecule (Ac) interacts with Cat^+^, H^+^ is transferred to Ac giving rise to a reactive carbene and proton disolvate Ac-H^+^-Ac. Thus, the alleged high reactivity of vinyl cations seems to be an exaggeration.

## 1. Introduction

Carbocations as positively charged particles are strong electrophiles and may (1) react with a nucleophile thus yielding adducts (SN1 reaction), (2) act as a protonating agent turning into highly active species that enter into secondary reactions, or (3) get rearranged into other carbocations [1,2,3]. These reactions cannot be studied in liquid superacids, where carbocations have so far been mainly investigated by NMR [2], because nucleophiles are inevitably protonated under these conditions. Therefore, experimental studies on carbocation/nucleophile interactions are scarce.

The reactivity of unsaturated carbocations having C=C and C≡C bonds is more difficult to study than that of saturated ones. As a class of reactive intermediates, they have been the subject of extensive theoretical and experimental research in the past five decades [4,5,6,7,8,9,10,11,12]. Nonetheless, most reactivity studies have been focused on solvolysis reactions where the reactive vinyl cation is intercepted by heteroatom-containing solvent molecules [4,5,13], and these reactions have not been analyzed in detail. It follows from these works that vinyl cations are highly reactive and, therefore, uncontrollable intermediates. This point of view has been refuted by Mayr and coworkers [14], who found that the vinyl cation is even less reactive than diarylcarbenium cations, some of the most stable trisubstituted cations. An overestimation of the reactivity of the vinyl cation, as follows from the research on solvolysis reactions, is also evidenced by high stability of vinyl cation salts in solutions in dichloromethane, from which they can be isolated into a crystalline phase, which has made it possible to study them by X-ray diffraction [15,16,17]. Recently, the reason for the apparent high reactivity of the C_6_H_5_CH_2_^+^ benzyl cation toward such a nucleophile as benzene was established: it protonates benzene, turning it into a highly reactive carbene, $C_{6}H_{5}\ddot{C}H$, which enters into a secondary reaction with the available carbocation [18].

We are not aware of reports about reactions of vinyl carbocations with the simplest oxygen-containing nucleophiles L. It would be expected that such reactions would proceed by the mechanism of both attachment of the nucleophile and its protonation, with the formation of proton disolvates L-H^+^-L as well, which have been studied by X-ray diffraction and IR spectroscopy (for L = H_2_O, Et_2_O, benzophenone, nitrobenzene, tetrahydrofuran, and others) [19,20].

In this work, we examined the interaction of unsaturated vinyl-type carbocations C_3_H_5_^+^ and C_4_H_7_^+^ (as carborane salts of the CHB_11_Cl_11_^−^ anion) with the simplest nucleophiles: water, alcohols, and acetone. The reaction products that crystallized were studied by X-ray diffraction and IR spectroscopy. We chose carborane CHB_11_Cl_11_^−^ as the counterion (hereafter denoted as (Cl_11_^−^), Figure 1) because of its exceptionally high stability at low basicity [21].

## 2. Results

The bulk amount of salts of cations C_3_H_5_^+^ and C_4_H_7_^+^ can be easily obtained by adding to the acid H(Cl_11_) powder such a small amount of 1,2-dichloropropane or 1,2-dichloro-2-methylpropane, respectively, with the sample remaining powdery. The quality of the resulting salts can be controlled by IR spectroscopy (their IR spectra should coincide with the reference spectra of salts [15,16], indicating the absence of impurities). These salts are soluble in pentafluorobenzene only in the presence of small amounts of water. Storage of such solutions for 1 day under ambient conditions led to a release of colorless crystals from it. The solubility of the salts and the yield of crystals increased with an increase in the content of water. X-ray diffraction analysis of crystals isolated from C_4_H_7_^+^(Cl_11_^−^) solutions showed that they are a salt of proton disolvate L−H^+^−L formed by two alcohol molecules L, representing 1-hydroxy-2-methylpropene (Figure 2).

The O⋅⋅⋅O distance in the OH^+^O moiety (2.420 Å) is typical for proton disolvates with very strong H-bonds [19,20]. The position of the bridging proton was determined by means of an electron density difference map (short O-H distance of 1.14 Å). Bond lengths and bond angles of the cation, which we will designate as I*a* (Figure 3), are given in Table 1 and Figure 2. Four carbon atoms C1-C4 and an oxygen atom of the alcohol molecule are in the same plane, and their CCC and CCO angles are close to 120°, i.e., C1 and C2 atoms have sp^2^ hybridization and, therefore, are double bonded. The C-O∙∙∙O angle is 118(3)°, which means that the O atom also has sp^2^ hybridization and belongs to the alcohol OH group. The C=C double bond length of 1.286 Å was determined as the average of two isomeric alcohol molecules (Appendix A). It is shorter than that in molecular C=C(–OH) fragments of enol tautomers (1.362 Å) [22]. The C−O distance of 1.252 Å is also slightly shortened compared to that in a single C–O bond in the same (C=)C–OH fragment (1.333 Å) [22].

The IR spectrum of the crystals is characteristic of proton disolvates: it contains an intense absorption pattern of the OH^+^O group, consisting of three broad bands at 905, 1297 and 1552 cm^−1^ (Figure 4, red). The band at 905 cm^−1^ belongs to ν_as_(OH^+^O), and bands 1297 and 1552 cm^−1^ to mixed stretching and bending vibrations of the OH^+^O group [20]. The shape of the band at 905 cm^−1^ is distorted by the resonance effect leading to so-called transparent windows (Evans holes) [23], which appear as dips in the spectrum at 855 and 640 cm^−1^.

The bands of CH stretching vibrations of CH_3_ groups of cation I*a* (Figure 4, inset) are easily interpreted because they are similar to those of CH_3_ groups of acetone in the previously studied disolvate Acetone−H^+^−Acetone [24], (Table 2). A weak band at 3048 cm^−1^ may belong to the stretching vibration of the =C-H bond, the C atom of which is adjacent to the OH^+^O group.

There are more uncertainties in the interpretation of the C-O and C=C stretching vibrations. It has been found that absorption bands of C−O stretching vibrations of alcohol molecules directly bound to H^+^ in proton disolvates are so strongly broadened and weakened in intensity that they are not detectable in IR spectra [24]. Two bands at 1560 and 1683 cm^−1^ (Figure 4) are in the expected frequency range of C=C stretch vibrations [15,16,17] and can be attributed to them. The fact that the position of the bridging proton is determined by X-ray diffraction means that its two-well potential has a high enough potential barrier for the proton to be at the bottom of one of the wells for a sufficiently long time (at the time scale of IR spectroscopy) in order to demonstrate (in an IR spectrum) the non-equivalence of two alcohol molecules in I as two C=C stretching frequencies. One of them at 1560 cm^−1^ is close to the frequency (1555 cm^−1^) of the cation in contact ion pair (CH_3_)_2_C=C^+^−H…(Cl_11_^−^). Therefore, it can be assumed that the frequency 1560 cm^−1^ belongs to the isobutylene alcohol molecule, which has a shorter O^+^−H bond (1.14 Å) and imitates a protonated alcohol molecule solvated by the second L alcohol molecule (CH_3_)_2_C=CH−OH−^+^H…L, which is less influenced by the positive charge and has an increased C=C stretch frequency: 1683 cm^−1^. The CC and CH stretch frequencies are summarized in Table 2.

It follows from the obtained results that disolvate Ia is generated by the interaction of the C_4_H_7_^+^ carbocation with water molecules, as shown in Figure 1. The weak spectrum of the H_3_O^+^ cation [25], which should arise simultaneously, actually manifests itself in the IR spectrum of the viscous phase, which precipitates concurrently with the crystalline phase.

X-ray diffraction analysis of crystals—grown from a solution of the C_3_H_5_^+^(Cl_11_^−^) salt under the same conditions under which crystals of the salt of Ia were obtained—showed partially disordered structure. This property does not allow to see in detail the entire structure of the cation but enables us to determine its similarity with cation Ia. The similarity is confirmed by the identity of the IR spectrum of the crystals to that of the salt of disolvate Ia (Figure 4), which means that the crystals growing from solutions of C_3_H_5_^+^(Cl_11_^−^) are proton disolvates L_2_-H^+^-L_2_, where L_2_ is CH_3_CH=CHOH. Hereafter, they will be referred to as cations Ib (Figure 2).

The solubility of salts of cations C_3_H_5_^+^ and C_4_H_7_^+^ in carefully dehydrated pentafluorobenzene is very low, and crystals cannot be grown from them. To increase the solubility of the salts, 1 vol% acetone was added to pentafluorobenzene containing trace amounts of water. This approach helped to obtain a solution with a heightened salt content and a reduced H_2_O/cation^+^ molar ratio. Keeping this solution over hexane vapor for 1 to 2 weeks led to the appearance of crystals. The X-ray diffraction analysis of the crystals obtained from solutions of the C_4_H_7_^+^(Cl_11_^−^) salts revealed that they contain the C_4_H_7_^+^ cation solvated by one molecule of 1-hydroxy-2-methylpropane (Figure 5). The crystal lattice does not have an acetone molecule but contains one solvent molecule, C_6_HF_5_, per two salt molecules. The C_4_H_7_^+^ cation is disordered over two positions differing in the location of the C2A atom slightly above or below the plane of three atoms: C1, C3, and C3A (Figure 5a and Appendix A). The C1-C2A∙∙∙O1 and C5-O1∙∙∙C2A angles are 137° and 121°, respectively (Figure 5b, Table 3), which means sp^2^ hybridization of C2A and O1 atoms, i.e., an H atom is attached to each of them. The C∙∙∙O distance is 2.470 Å, which matches the maximum allowable distance between the O atoms of the symmetric O∙∙∙H^+^∙∙∙O moiety in proton disolvates (2.47 Å for the most unstable proton disolvates obtained: H^+^(nitrobenzene)_2_) [19]. Therefore, with a high probability, a bridging proton is located between the C2A and O1 atoms, forming a short, strong, and low-barrier double-well H-bond, although compounds containing asymmetric moiety (C)C−H^+^∙∙∙O with a strong H-bond are currently unknown. The short distance (C2A)H∙∙∙O1 of 1.97 Å, also points to the presence of a strong H- bond. The significant difference of the C2A-H∙∙∙O1 angle at 111° from the optimal one at 180° may be partly due to the inaccuracy of determining the localization of the H atom by the calculation method. The question of the presence of a strong H-bond with double-well proton potential is discussed below when the IR spectra of these crystals are examined. The C_4_H_7_^+^·OHC_4_H_7_ cation under consideration is hereafter denoted as IIa (Figure 2).

X-ray diffraction analysis of crystals obtained from a solution of C_3_H_5_^+^(Cl_11_^−^) in C_6_HF_5_ + 1% acetone indicated that they contain the hydrocarbon cations and C_6_HF_5_ inclusion molecules of the solvents with significantly disordered C and F-atoms. We failed to localize the highly disordered structure of the cation. For this reason, we do not present or discuss these X-ray diffraction data. Nonetheless, they provide some useful information. For instance, in the crystal lattice, alcohol molecules C_3_H_5_OH with reliably fixed C and O atoms were identified, possibly indicating the solvation of the cation with the C_3_H_5_OH molecule in the same way as the C_4_H_7_OH molecule solvates the C_4_H_7_^+^ cation in adduct II*a*. That is, we can assume the emergence of cationic adduct C_3_H_5_^+^·C_3_H_5_OH, similar to IIa, and designate it as IIb (Figure 2). The IR spectra of crystals containing IIa and IIb adducts are identical (Figure 6), which confirms that IIb is C_3_H_5_^+^·C_3_H_5_OH.

In the IR spectra of cationic adducts IIa and IIb, specific absorption of a strong C–H^+^∙∙∙O H-bond should be present; however, this type of strong H-bond has not yet been found. Well-studied strong hydrogen bonds are symmetric O∙∙∙H^+^∙∙∙O or N∙∙∙H^+^∙∙∙N H-bonds in proton disolvates L_2_H^+^ with double- well proton potential separated by a low potential barrier that transforms it into flat-bottom potential for vibrational transitions. In these cases, the proton vibrations appear in the IR spectra as an intense and broad absorption pattern with a maximum at 850–1000 cm^−1^ [25], as observed in the spectra of cations Ia and Ib (Figure 4). In proton disolvates with an asymmetric moiety, for example, N−H^+^∙∙∙O, the bottom of the double-well potential is asymmetric and the maximum of the broad and intense absorption shifts to 1400–1700 cm^−1^ [25]. In the spectra of the analyzed adducts IIa and IIb, a broad band at 1630 cm^−1^ is observed, as is absorption in the region of 1200–1500 cm^−1^, which is not described by a single Gaussian (Figure 6, inset). They correspond fairly well to the asymmetric X_1_−H^+^∙∙∙X_2_ fragment, in our case =C−H^+^∙∙∙O.

The C=C bond of cations C_3_H_5_^+^ and C_4_H_7_^+^ is affected by the bridging proton, and this effect should specifically influence the absorption of its stretch vibrations. For example, it has been found [24] that if a single C-O bond is attached to a bridged proton its absorption band broadens and decreases in intensity so much that it is undetectable in the IR spectrum. If the C=O bond is attached to H^+^, as in proton disolvate Acetone-H^+^-Acetone (discussed below), it still appears in the spectrum as a weak-to-moderately intense broadened band [24]. Therefore, absorption of the C=C stretch vibration of C_3_H_5_^+^ and C_4_H_7_^+^ may look like a weak band at 1633 cm^−1^ (Figure 6). It is higher in frequency than νC=C at ~1560–1590 cm^−1^ in the contact ion pairs (CIPs) formed with the (Cl_11_^−^) anion in solutions and in a solid phase [15,16,17]. This means that the interaction of C_3_H_5_^+^ and C_4_H_7_^+^ cations with an alcohol molecule in II is stronger than the interaction with the (Cl_11_^−^) anion in the CIP.

The C=C band of the alcohol molecule in adducts IIa and IIb is subject to a weaker influence of the bridging proton. Its C=C stretching vibration is observed at 1704 cm^−1^ (Figure 6). It is a single band for adduct IIb, whereas in the spectrum of IIa it is split into two components at 1715 and 1694 cm^−1^ of equal intensity (Figure 6, inset). Obviously, in the crystal lattice of the II*a* adduct salt, there are two weakly nonequivalent C_4_H_7_OH molecules.

In the CH stretch frequency region, the three bands at 2961, 2932 and 2874 cm^−1^ can be unambiguously interpreted as vibrations of the CH_3_ group (Table 2). The IR spectra of the crystals also contain absorption bands of the captured pentafluorobenzene molecule. They are easily identified because of the finding that when a crystal is crushed on an ATR accessory and its crystal lattice is destroyed, pentafluorobenzene is released and slowly evaporates when the sample is kept in a glove box atmosphere. Therefore, the intensity of its absorption decreases over time. The most intense C_6_F_5_H bands in Figure 6 are marked with asterisks.

The band at 3370 cm^−1^ may belong to OH groups of the C_4_H_7_OH alcohol molecule, which are engaged in weak H bonds with the (Cl_11_^−^) anion.

The solution from which crystals of the salt of II*b* grew was kept in a sealed ampoule, and after a few days a small number of tiny crystals grew from it. It was possible to find one crystal of sufficient size for X-ray analysis. It turned out that this was a salt of the monohydrate of the propylene cation (C_3_H_5_^+^∙OH_2_)(Cl_11_^−^) (Figure 7). The C_3_H_5_^+^∙OH_2_ species has two localizations in the unit cell with slightly different positions of C and O atoms, but they can be distinguished (Figure 7a). Similarly, an anion can be disordered over two positions, as indicated by the presence of electron density peaks in the difference map in the region of the anion. Nonetheless, we failed to localize the second position of the anion. This may be the reason for substantial deterioration of the R_f_ factor, but does not interfere with the determination of coordinates of the C and O atoms of the cation with accuracy sufficient to establish the topology of the cation qualitatively and its main geometric parameters (Table 4).

The main feature of the structure of the C_3_H_5_^+^∙OH_2_ cationic adduct, which is denoted below as III*b* (Figure 2), is as follows. The H_2_O molecule is attached to the C_3_H_5_^+^ cation through the O atom in the direction perpendicular to the C=C double bond at a distance of 2.32 Å, which is very close to the average Na···O(H_2_) distance of 2.333 Å in the first hydration shell of hexahydrate Na(OH_2_)_6_ as determined by X-ray diffraction analysis for 13 structures (retrieval was made according to the Cambridge structural data base, ConQuest 2021.3.0) [26]. Therefore, the nature of the interaction of the H_2_O molecule with the C=C^+^ bond of the vinyl cation is similar to that of the interaction of water molecules with the alkali metal cation in its first hydration shell.

Unfortunately, the insufficient amount of the obtained crystals of (C_3_H_5_^+^∙OH_2_)(Cl_11_^−^) did not allow us to register their IR spectrum. Nevertheless, the spectra of these compounds have been obtained by us earlier, when we characterized crystalline salts C_3_H_5_^+^(Cl_11_^−^) and C_4_H_7_^+^(Cl_11_^−^) by X-ray diffraction and IR spectroscopy [15,16]. In the IR spectra of individually selected single crystals of these salts subjected to X-ray analysis, no traces of water absorption were found. On the other hand, during the recording of IR spectra of an aggregate of small crystals that arose simultaneously with larger ones, a weak spectrum of water molecules was observed (Figure 8). Its frequencies ν_as_, ν_s_ and δ at 3612, 3544, and 1606 cm^−1^, respectively, are similar to those of monomeric molecules dissolved in organic solvents or hydrated alkali metal cations (Table 5). This result was surprising and could not be explained. Now it is clear that the observed bands belong to non–H-bonded water molecules of monohydrates (C_4_H_7_^+^∙OH_2_)(Cl_11_^−^) and (C_3_H_5_^+^∙OH_2_)(Cl_11_^−^) designated subsequently as IIIa and IIIb (Figure 2), which are formed and co-crystallize with salts of anhydrous vinyl cations. They are typical of the H_2_O molecules that hydrate the H_3_O^+^ cation in H_3_O^+^(H_2_O)_3_(Cl_11_^−^) crystals [27], or alkali metal cations (Table 5). Such mostly ionic interaction Cat^+^···OH_2_ leads to a slight decrease in the frequencies of OH stretching, compared to those of the dissolved monomer molecule. In adducts III*a/b*, OH stretching frequencies are somewhat lower than those in hydrates of alkali metal cations and H_3_O^+^; hence the strength of bond Cat^+^···OH_2_ in III*a/b* is higher.

As mentioned above, crystals of compounds IIa and IIb were obtained from saturated solutions of vinyl cation salts in C_6_HF_5_ containing 1 vol% acetone. It could be expected that after an increase in the acetone content, crystals containing acetone could be obtained. Incubation of saturated solutions of the C_3_H_5_^+^(Cl_11_^−^) or C_4_H_7_^+^(Cl_11_^−^) salts in C_6_HF_5_ + 5% acetone over hexane vapor gave rise to needle-like crystals. X-ray diffraction analysis indicated that this was proton disolvate salt Ac-H^+^-Ac (IV), where Ac is the acetone (Figure 2 and Figure 9a). The same salt was obtained in a different way by reacting the H(Cl_11_) acid with acetone vapor and its subsequent dissolution in dichloromethane. Slow evaporation of the solution in the glove box led to the formation of crystals. Their X-ray diffraction analysis showed that this was also proton disolvate salt (Ac)_2_H^+^(Cl_11_^−^) but with other crystal lattice parameters, that is, it is a different polymorph (Figure 9b). The O⋅⋅⋅O distance in cations for both polymorphs is almost the same, 2.429 and 2.423 Å. Selected geometric parameters of IV and IV′ are given in Table 6.

The IR spectra of the (Ac)_2_H^+^ cation (Appendix A) match its known IR spectra [24,28]; interpretation of the spectrum is given in ref. [28].

## 3. Discussion

The results make it possible to establish the sequence of the interaction of vinyl cations with water molecules. Initially, an H_2_O molecule is attached to the C=C^+^ bond of cation RCH=C^+^H or R_2_C=C^+^H (where R is CH_3_) via the O- atom (Equation (1), in it and in subsequent equations, the R_2_C=C^+^H cation is used).



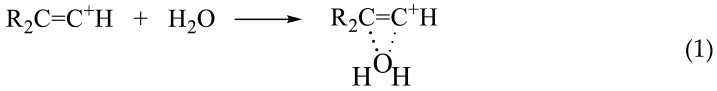



The nature of this bond is close to that of bonds formed by water molecules with alkali metal cations during their hydration or with H_3_O^+^ in crystal salts of H_3_O^+^·3H_2_O, i.e., the bond is strongly ionic.

With an increase in the concentration of the salt of cationic adduct R_2_C=C^+^H···H_2_O in solutions, the self-association of ion pairs increases (which is typical for salts of carbocations in solutions [29]). This enhances the contact interaction of H_2_O with the cation and promotes the transfer of a proton to a water molecule with the formation of H_3_O^+^ and cationic adduct II:



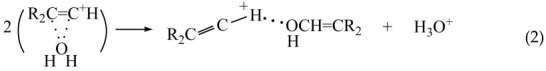



With a further increase in the contents of water and of the carbocation salt, the concentration of the resulting adducts II and of the H_3_O^+^ cation increases. The latter can protonate an alcohol molecule that is more basic than H_2_O, thereby producing a proton disolvate and regenerating cation R_2_C=C^+^ H (Equation (3)),


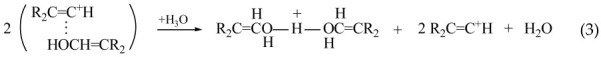
which next interacts with water, closing the cycle. The alcohol molecule of adduct II can also be protonated directly by cation III, thus by passing the stage of H_3_O^+^ formation:



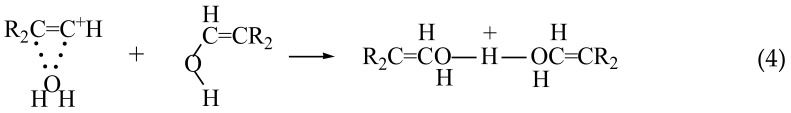



The studied cationic adducts enable one to trace the change in the nature of the interaction of vinyl cations with basic molecules of H_2_O, alcohol, and acetone as their basicity increases. The H_2_O molecule is attached to the C=C bond of the vinyl cation (Figure 2, III) with a strength similar to that of an H_2_O molecule attached to an alkali metal cation or to the H_3_O^+^ cation in H_3_O^+^(H_2_O)_3_. As the basicity of the O atom increases from H_2_O to alcohol molecule, its interaction with the cation shifts to the H atom, thereby producing a strong H-bond, with a partially covalent character [30], and an increase in the C⋅ ⋅ ⋅ ⋅ ⋅O distance to 2.47 Å (Figure 2, II). In this case, the asymmetric double-well potential of the bridging proton has a deeper minimum at the C atom. A further increase in the basicity of the oxygen atom (acetone molecule) causes a shift of the minimum of the double-well potential to the O atom with a high probability of proton transfer to the acetone molecule. The loss of a proton by the vinyl cation results in an extremely reactive carbene molecule (with C=C: as the active site), which then reacts with the components of the mixture, forming non-crystallizing products (wax phase). The released protonated acetone adds a second acetone molecule thereby generating proton disolvate IV.

Thus, the interaction of the vinyl cation with nucleophile L proceeds: (1) through its addition to a charged double bond (if L = H_2_O) or H-bonded to the =C-H group (if L = alcohol molecule), and (2) through the protonation of L with the transition of the vinyl cation to the neutral carbene molecule. The second mechanism of interaction was proposed in ref. [31] and proved for the benzyl carbocation [32].

The finding that adducts II and III with strongly ionic Cat^+^−L interaction exist is surprising. They can exist if the basicity of L slightly exceeds that of counterion (Cl_11_^−^). This means that vinyl cations behave like rather chemically inert particles, contradicting predictions of quantum chemical calculations. For example, if the crystallographic structures of adducts II and III are optimized, then the covalent interaction of the cation with the H_2_O or alcohol molecule results in the emergence (in the case of C_4_H_7_^+^) of molecules of protonated isobutenyl alcohol and diisobutenyl ether, respectively (Figure 10), with a significant gain in energy. 

For example, if we compare energies of the structures calculated at the UB3LYP/6- 311++G(d,p) level of theory: (1) energy with optimized H atomic coordinates and fixed coordinates of C and O that are equal to the coordinates that follow from the X-Ray data for IIa, and (2) energy with fully optimized structure (including the C, O and H atomic coordinates), energies (1) and (2) are related as 69.71 and 0 kcal/mol with the transformation of structure IIa into protonated diisobutenyl ether. The environment cannot have a stronger stabilizing effect on the IIa structure because the purely ionic interaction of C and O atoms with Cl atoms of the anionic environment is weak (C···Cl and O···Cl distances exceed the sum of van der Waals atomic radii). Thus, it follows from quantum chemical calculations that adducts II and III should not exist, which contradicts the experimental findings. Therefore, the use of quantum chemical calculations requires caution in studies on mechanisms of the interaction of vinyl cations with neutral molecules.

## 4. Materials and Methods

The salts of vinyl cations C_3_H_5_^+^ and C_4_H_7_^+^ were obtained as described previously [15,16,17]. The pentafluorobenzene (Sigma-Aldrich, Saint Louis, MO, USA) used as a solvent was thoroughly dried with molecular sieves and was not purified further.

All sample handling was carried out in an atmosphere of argon (H_2_O, [O_2_] < 0.5 ppm) in a glove box. ATR IR spectra were recorded on a Shimadzu IRAffinity-1S spectrometer housed inside the glove box in the 4000−400 cm^−1^ frequency range using an ATR accessory with a diamond crystal. The spectra were processed in the GRAMMS/A1 (7.00) software from Thermo Scientific, Waltham, MA, USA.

X-ray diffraction data were collected on a Bruker Kappa Apex II CCD diffractometer using φ,ω-scans of narrow (0.5°) frames with Mo Kα radiation (λ = 0.71073 Å) and a graphite monochromator at temperature 200 K. The structures were solved by direct methods with the help of SHELXT-2014/5 [33], and refined by the full-matrix least-squares method against all F2 in an anisotropic-isotropic (for H atoms) procedure using SHELXL-2018/3 [33]. Absorption corrections were applied by the empirical multiscan method using SADABS software [34]. Hydrogen atom positions were calculated using the riding model. The hydrogen atom positions for OH -groups were located by means of a difference Fourier map. The crystallographic data and details of the refinements for all structures are summarized in Appendix A. We were unable to obtain good single crystals of the (C_3_H_7_^+^⋅ H_2_O)(Cl_11_^−^) salt (of adduct III) for X-ray diffraction analyses. Nonetheless, we conducted an X-ray diffraction experiment and localized non-hydrogen atoms, but could not refine the structure to obtain a good R-factor. Crystals of the salt of cationic adduct II contain strongly disordered C_6_HF_5_ solvent molecules. We were not able to find their atomic coordinates. The accessible volume of free solvent molecules in these crystals, as determined by routine PLATON analysis, was 15.0% (846 Å^3^). The highly disordered C_6_HF_5_ molecules occupying this volume could not be modeled as a set of discrete atomic positions. We employed the PLATON/SQUEEZE procedure to calculate the contribution to the diffraction from the solvent region, and thereby produced a series of solvent-free diffraction intensities. The independent part of the unit cell of salt [(CH_3_)_2_C=CHOH]_2_H^+^(Cl_11_^−^) contains two anions and two cationic proton disolvates I. In Table 1, the geometry of disolvate I is given as an average over two solvate molecules (CH_3_)_2_C=CHOH. A similar averaging of geometric parameters over two independent positions was performed for adduct III and proton disolvates IVa and IVb (Table 4 and Table 6).

CCDC 2223519, 2223520, 2223521, 2223522 and 2223523 contain the supplementary crystallographic data for this paper. These data can be obtained free of charge from The Cambridge Crystallographic Data Center at http://www.ccdc.cam.ac.uk/data_request/cif (accessed on 9 January 2023).

The obtained crystal structures were analyzed for short contacts between non-bonded atoms using PLATON [35,36] and MERCURY software packages [37].

## 5. Conclusions

Vinyl cations C_3_H_5_^+^ and C_4_H_7_^+^ (Cat^+^) in solutions of their salts in dichloromethane and C_6_HF_5_ interact with O-containing nucleophiles as follows.

An H_2_O molecule attaches to the C=C bond of Cat^+^ in a similar manner to the hydration of alkali metal cations, thereby yielding Cat^+^·H_2_O adducts with a strongly ionic bond. Its strength only slightly exceeds the strength of the interaction of the (Cl_11_^−^) anion with the vinyl cation in contact ion pairs Cat^+^(Cl_11_^−^).

With an increase in the content of Cat^+^·H_2_O adducts in solutions, the adduct self-associates and interacts with a transfer of a proton to one water molecule and attachment of the second one to the C=C bond, thus forming H_3_O^+^ and an alcohol molecule, respectively.

The alcohol molecule interacts predominantly with the H atom of the C=C^+^−H moiety of the vinyl cation, thereby producing a proton disolvate with strong asymmetric H-bond =C−^+^H···O having double-well proton potential with a deeper minimum near the C atom.

A further increase in the water content in the solutions leads to complete conversion of vinyl cations into alcohol molecules with the formation of symmetric proton disolvates LH^+^L containing strong and partially covalent O−H^+^−O hydrogen bonds [30] and H_3_O^+^ cations.

The interaction of the vinyl cations with acetone molecules, which are more basic than H_2_O or alcohol molecules, causes the formation of only symmetrical proton disolvates, LH^+^L, in the absence of acetone-containing cationic adducts. The vinyl cation is converted into carbene containing a highly reactive C=C: moiety.

Summing up, we can say that the interaction of the vinyl cation with base L proceeds through two mechanisms: via the formation of adducts (SN1 reaction), and via the mechanism where vinyl cation acts as a protonating agent. When the basicity of L is close to that of a single water molecule, L attaches to the double C=C bond thereby producing an adduct. As the basicity of L increases, the interaction with the C=C^+^−H moiety of the vinyl cation strengthens and is shifted to the H atom, thus forming a solvate having a strong asymmetrical =C−^+^H···O hydrogen bond. Further strengthening of the basicity of L leads to the transfer of a proton to L and to the emergence of the eventually symmetric LH^+^L cation. The loss of a proton by the vinyl cation converts it into a neutral reactive carbene molecule containing a C=C: moiety.

The formation of adducts with water and alcohol molecules by vinyl cations is unexpected, because according to quantum chemical calculations, they are energetically unfavorable and should not exist.

The very existence of these adducts means that the alleged high reactivity of vinyl carbocations is an overestimation.

## Data Availability

Not applicable.

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
