# Peer review of "Interaction of Vinyl-Type Carbocations, C3H5+ and C4H7+ with Molecules of Water, Alcohols, and Acetone"

_molecules, 2023, doi:10.3390/molecules28031146_

Round 1

Reviewer 1 Report

The manuscript of Stoyanov et al. presents the study on the activity of vinyl cations with nucleophiles, which is  supported by the analysis of several crystal structures and some computational studies.

Though the manuscript reports a comprehensive study of  the reactivity of two different vinyl cations, is well supported by experimental details, and the importance of the reported results is generally high, the manuscript can be significantly improved and made much more easier to read and understand. Iit lacks is, first, at least a couple of sentences / a scheme regarding the preparation of these carbocations in the beginning of the manuscript. These species are the key of the current study, and just referring to the previous publications is not appropriate.  Second, it should include an overview Figure / Scheme clearly showing all the molecular structures discussed through the manuscript. Just saying "Cation IIa" or something similar in the text makes it confusing and difficult to understand. Please, add such an "overview" figure somewhere in the beginning of the paper.

Several other issues to be considered:

- Did the authors try to perform the search of the transition states for the interconverting cationic species? It may happen that though the initial specie is much higher an energy, a high transition state may preclude the reaction kinetically.

- Figure S1 can be moved to the main body of the paper from SI.

- In SI the authors say "will be published" for the figures S2 and S3 of the chlorinated adduct with C4H8Cl. This adduct appears out of blue and either requires more detailed explanation or should not be mentioned at all. If the authors still want to mention it, they may consider including its X-ray structure too.

Author Response

Response to Reviewer report

Reviewer 1

Comments and Suggestions for Authors

The manuscript of Stoyanov et al. presents the study on the activity of vinyl cations with nucleophiles, which is supported by the analysis of several crystal structures and some computational studies.

Though the manuscript reports a comprehensive study of  the reactivity of two different vinyl cations, is well supported by experimental details, and the importance of the reported results is generally high, the manuscript can be significantly improved and made much more easier to read and understand. Iit lacks is, first, at least a couple of sentences / a scheme regarding the preparation of these carbocations in the beginning of the manuscript. These species are the key of the current study, and just referring to the previous publications is not appropriate. 

It's done. A couple of sentences have been added to the beginning of section 2. Results

Second, it should include an overview Figure / Scheme clearly showing all the molecular structures discussed through the manuscript. Just saying "Cation IIa" or something similar in the text makes it confusing and difficult to understand. Please, add such an "overview" figure somewhere in the beginning of the paper.

We have added an "overview" Figure 3 with a schematic representation of all the carbocations studied in the work.

Several other issues to be considered:

- Did the authors try to perform the search of the transition states for the interconverting cationic species? It may happen that though the initial specie is much higher an energy, a high transition state may preclude the reaction kinetically.

No. We investigated only the initial and final states, which can be proved by X-ray diffraction analysis. The study of transition states requires the use of other research methods, and this is another work that cannot be performed within the framework of this work.

- Figure S1 can be moved to the main body of the paper from SI.

We did it.

- In SI the authors say "will be published" for the figures S2 and S3 of the chlorinated adduct with C4H8Cl. This adduct appears out of blue and either requires more detailed explanation or should not be mentioned at all. If the authors still want to mention it, they may consider including its X-ray structure too.

Figures S3 and S4 have been removed from the SI, the necessary changes have been made in the text.

Reviewer 2 Report

This is a very interesting study on interactions between carbocations and water. This process is described as consisting of the following stages; the formation of adduct (CH...O linkage) and next the proton movement from the carbocation to the water molecule, the H3O+ cation is finally formed.

The latter process is described on the basis of X-ray crystal structure data and IR-spectroscopy.

This article deserves to be published almost as it stands; there are only few minor suggestions from my part;

- the edition error: the numeration of sections of the manuscript is strange (number 1 appears for majority of sections),

- the experimental results are compared with theoretical ones sometimes; few additional experimental and theoretical studies on the related systems may be taken into account by the authors; as for example;

1.          J. Phys. Chem. A 2008, 112, 1897–1906.

2.      J. Phys. Chem. A 2007, 111, 13537-13543.

also important studies of Perrin may be mentioned.

Author Response

Reviewer 2.

Comments and Suggestions for Authors

This is a very interesting study on interactions between carbocations and water. This process is described as consisting of the following stages; the formation of adduct (CH...O linkage) and next the proton movement from the carbocation to the water molecule, the H3O+ cation is finally formed.

The latter process is described on the basis of X-ray crystal structure data and IR-spectroscopy.

This article deserves to be published almost as it stands; there are only few minor suggestions from my part;

- the edition error: the numeration of sections of the manuscript is strange (number 1 appears for majority of sections).

Done (this happened automatically when the text of the article was transferred to the template in the editorial office)

- the experimental results are compared with theoretical ones sometimes; few additional experimental and theoretical studies on the related systems may be taken into account by the authors; as for example;

  1. J. Phys. Chem. A 2008, 112, 1897–1906. [G. E. Douberly, A. M. Ricks, B. W. Ticknor, W. C. McKee, P. v. R. Schleyer, and M. A. Duncan, Infrared Photodissociation Spectroscopy of Protonated Acetylene and Its Clusters] https://doi.org/10.1021/jp710808e
  2. J. Phys. Chem. A 2007, 111, 13537-13543. [Sławomir Janusz Grabowski, π−H···O Hydrogen Bonds: Multicenter Covalent π−H Interaction Acts as the Proton-Donating System] https://doi.org/10.1021/jp076990t

About the use of references (1) and (2). Our study focuses on how C3H5+ and C4H7+ cations interact with oxygen-containing nucleophiles. Therefore, at the beginning of the Introduction, we referred only to three review papers [1-3] on unsaturated carbocations with C=C and CºC bonds, which also contain some information on their reactivity. Also refer to works in which the state of C3H5+ and C4H7+ cations in solid salts was studied. A review is given of works that deal with the topic of the reactivity of specific carbocations.

Work (1), as well as other works by M. A. Duncan et al., we referred to earlier in our articles on the study of carbocations themselves (our references [14-16] and others). In (1), only the state of protonated acetylene and its interaction with other acetylene molecules in the gas phase was studied. This is a different research topic compared to the current one, and we do not know where in our article we can refer to it, so that this reference would be appropriate. We worked with C2H3+{Cl11-} salts, but the paper is not ready for publication. It is there that reference (1) is used, where it is in the subject. The state of the C2H3+ cation (its IR spectra) in solid salts and in the gas phase is different.

In (2), the interactions π-H×××O, O-H×××π, and π-H×××π, which can be classified as hydrogen bonds, were analyzed using quantum chemical calculations. None of the cationic adducts analyzed by us contains H-bonds of this type (only С+-H×××O H-bonds are available). They are very different in nature to be comparable. Therefore, we do not know how to refer to the article (2) and discuss it in our work.

also important studies of Perrin may be mentioned.

Reviewer 3 Report

The authors present an experimental spectroscopic study related to the characterization of structure and intermolecular contacts. Values of vibrational modes, angles and bond lengths are discussed. Of course that the experimental nature of this current study is worthy and notable, although, in my point of view, the authors could have carried out some theoretical calculations, which are accessible to everyone. This endeavour would give support to the experimental data as well as new horizons could be rise. Let's take an specific point widely known, the hydrogen bond formation is closely related to the advent of shifted frequencies to red or blue. This is not experimentally explored, although reiterating, if theoretical calculations have bee done, this kind of analysis would be easily performed because would consist in a comparison between the oscilattors of the monomers and supermolecule. Indeed, the use of theoretical calculations would strengthening the manuscript, e.g., the structure 1a (Figure 1), the hydrogen bond distance of 1.28 makes it a very strong interaction, and thereby, it could be characterized as partial or total intermolecular covalent if theoretical calculations have been done. As a matter of fact if the authors have decided to go ahead exclusively with the experimental evaluation, regardless the results, e.g. those gathered in Table 1, all are poorly explored.

Amazingly this kind of project is no routinely approached, by which there is a certain lack of publications. Even so, the list of references is relatively old, and some more recent publications shall be cited by the authors, namely as:

ttps://doi.org/10.1021/acs.chemrev.5b00484

10.1021/acs.orglett.0c01745

Moreover, the authors have pointed out and concluded regarding the existence of hydrogen bonds, although no reference about this topic has been cited. I suggest some article to be cited in the manuscript, such as:

https://doi.org/10.1002/anie.201002960

https://doi.org/10.1021/cr800346f

Author Response

Reviewer 3.

Comments and Suggestions for Authors

The authors present an experimental spectroscopic study related to the characterization of structure and intermolecular contacts. Values of vibrational modes, angles and bond lengths are discussed. Of course that the experimental nature of this current study is worthy and notable, although, in my point of view, the authors could have carried out some theoretical calculations, which are accessible to everyone This endeavour would give support to the experimental data as well as new horizons could be rise.

We wrote on page 12 that structures III and II, according to quantum chemical calculations [at the UB3LYP/6- 311++G(d,p) level of theory] should not exist, but should be transformed to protonated isobutenyl alcohol and diisobutenyl ether, respectively (Figure 10), with a significant gain in energy. Therefore, a more detailed quantum chemical study does not make sense to conduct. In our previous papers, we have shown ([1] ACS Omega 2021 6, 15834. DOI: 10.1021/acsomega.1c01297; [2] ACS Omega 2021 6, 23691. DOI: 10.1021/acsomega.1c01316; [3] Int. J. Mol. Sci. 2022, 23, http://doi.org/10.3390/ijms23169111), that quantum chemical calculations are poorly applicable to unstabilized carbocations with C=C and СºС bonds, on which the “+” charge is concentrated. The discrepancy between the calculated C=C stretch frequencies and the experimental ones is ~300 cm-1 [1,2}], and for СºС it reaches ~400 cm-1 [3], which is unacceptable. According to the calculations, the cation (СH3)2C=CH+ is unstable and must pass into chain carbocation, but it has been experimentally obtained and is thermally stable up to 150 °С [1]. Our work shows how it interacts with oxygen-containing nucleophiles while retaining its isomeric C-skeleton.  Theoretical calculations show good applicability only to stabilized vinyl cations in which the “+” charge is dispersed over the substituents. Therefore, in this work we did not use deeper theoretical calculations than we did. Such studies should be carried out by highly qualified specialists in the field of quantum chemistry.

Let's take an specific point widely known, the hydrogen bond formation is closely related to the advent of shifted frequencies to red or blue. This is not experimentally explored, although reiteratin, …

It is not clear why experimentally is not explored? We have one complex with strong hydrogen bond with C-H stretch at 1630 cm-1 (Figure 6, Table 2) and two proton disolvates I and IV with nasOHO ~ 900 cm-1. How can these frequencies be discussed in terms of red shift or blue shift? This kind of discussion is completely out of place.

…if theoretical calculations have been done, this kind of analysis would be easily performed because would consist in a comparison between the oscillators of the monomers and supermolecule. Indeed, the use of theoretical calculations would strengthening the manuscript, e.g., the structure 1a (Figure 1),…

Organic compounds with strong and very strong H-bonds were intensively studied 30-40 years ago. By the time we published our articles [18[ and [19], to which we refer here, these studies were considered completed and since then nothing new has appeared on the interpretation of the IR spectra of such compounds. Therefore, we see no reason to reproduce calculations that have been made long ago and are well known.

…the hydrogen bond distance of 1.28 makes it a very strong interaction, and thereby, it could be characterized as partial or total intermolecular covalent if theoretical calculations have been done.

Strong hydrogen bond is partially covalent. This has been proven a long time ago and we do not need to prove it again. We have added a ref. [30] suggested by a reviewer to the review paper “What Is the Covalency of Hydrogen Bonding? Chem. Reviews, 2011”. It makes no sense for us to reproduce this result.

 As a matter of fact if the authors have decided to go ahead exclusively with the experimental evaluation, regardless the results, e.g. those gathered in Table 1, all are poorly explored.

We have already answered above that we rely on completed studies of compounds with a strong H-bond.

Amazingly this kind of project is no routinely approached, by which there is a certain lack of publications. Even so, the list of references is relatively old, and some more recent publications shall be cited by the authors, namely as:

ttps://doi.org/10.1021/acs.chemrev.5b00484 (The Halogen Bond, Chem. Reviews, 2016)

The requirement to use this reference is confusing me. It is a large overview article "Halogen bond". There are no halogen bonds in the carbocation adducts studied by us. I don't know how and where to cite this work.

We have previously worked with chloronium cations, such as divinyl chloronium (C2H3)2Cl+ (Int. J. Mol. Sci. 2022, 23, Issue 16), R-Cl+-R (R = CH3, CH2Cl, C2H5, C3H7, published by us earlier) with a pronounced Halogen bond, and there this reference to the place.

10.1021/acs.orglett.0c01745 (Urea-Catalyzed Vinyl Carbocation Formation Enables Mild Functionalization of Unactivated C−H Bonds, Org.Lett. 2020).

We added it as reference 12 in Introduction.

Moreover, the authors have pointed out and concluded regarding the existence of hydrogen bonds, although no reference about this topic has been cited.

We do not agree with this. On page 2, when disolvate Ia is given, references [18, 19] are given (the words with very strong H-bonds are added). In [18](J. Am. Chem. Soc., 2002, 124, 13869), organic proton disolvates are investigated and a review of the literature at that time is given. In [19] (J. Phys. Chem .A. 2006, v.110, 12992) the H5O2+ cation as a proton disolvate is investigated and a detailed literature review is given on the IR spectra and the nature of the L-H+-L hydrogen bond. The nature of the H-bond of the symmetric O-H+-O fragment was also experimentally studied in detail. We do not give more references, since [18 and 19] collected all the literature available at that time. Since then, nothing fundamentally new has appeared in the literature on this topic. In the text, wherever the strong hydrogen bond is discussed, references are made to these two articles, as well as to Johannsen's work in book [24] on strong asymmetric hydrogen bonds X-H+…Y.

I suggest some article to be cited in the manuscript, such as:

https://doi.org/10.1002/anie.201002960

Reference to this work (A Bond by Any Other Name, Angew. Chem. Int. Ed. 2011, 50, 52-59) can only be given if we undertake to discuss the nature of the O-H+-O bond in I, IV and C-H+…O in II. But we do not have enough new data, in addition to those already known for these bonds, for such a discussion.

We are currently working on preparing an article for publication discussing the isobutylene carbocation solvated by the isobutylene molecule:

In it, the distance C1B….C1A between the C atoms of the cation and the alkene molecule is as short, 2.44 Å, as between the X atoms of the X-H+-X fragment in proton disolvates L-H+-L. This bond is formed due to the supply of p-e- density from the C1a=C2ca bond of the olefin to the C1b+ atom of the cation through the H1a atom. The H atom in this case plays the role of the electronegative atom directed to electropositive C atom of the carbocation. This can be called an inverse H-bond with respect to the H atom (if at all it can be called a hydrogen bond?). In this work, the use of the article ”A Bond by Any Other Name” in Angew. Chem. is very appropriate.

https://doi.org/10.1021/cr800346f (What Is the Covalency of Hydrogen Bonding? Chem. Reviews, 2011)

Was added it as ref.[30].

Reviewer 4 Report

            These are interesting results, but they are marred by frequent overinterpretation.  I strongly encourage the authors to scale back on the interpretations and focus on what the data document.

            I consider that the similarity of the distances of 2.32 and 2.333 Å is fortuitous and does not indicate that water interacts in similar fashion with C3H5+ and Na(OH2)5+ (whatever "similar" means).  What is the evidence that water interacts with the C=C of the cation, rather than with the center of positive charge?  That conclusion seems to come from Fig. 5, but why are there two oxygens in 5a, why is a carbon missing in 5b, and how do the neighboring {Cl11-} anions in the crystal (not shown) affect the structure?

            The claim in the Abstract for a (meaningless) "short, strong, low-barrier (SSLB) asymmetric H–bond" between the C-H and an alcohol is not supported anywhere in the document and should be removed.  Is the self-citation (Ref. 18) really the shortest O-O distance in a hydrogen-bonded species?

            The final conclusion is contradictory: "The formation of adducts ... by vinyl cations is unexpected, because according to quantum chemical calculations, they are energetically unfavorable and should not exist.  Thus, the alleged high reactivity of vinyl carbocations is an overestimation."  If quantum-chemical calculations imply that adducts of vinyl cations should not exist, whereas they do form, then the calculations are faulty, and vinyl cations are indeed reactive, as expected.

            Minor: (1) I recommend calling alcohol L an enol, to emphasize throughout the unusual nature of this alcohol.  (2) Do not call adduct III adduct A in the Abstract.  (3) Why is adduct III discussed after I and II even though it is the first formed?

Author Response

Reviewer 4

  These are interesting results, but they are marred by frequent overinterpretation.  I strongly encourage the authors to scale back on the interpretations and focus on what the data document.

We removed one of the proofs on interpretation (p. 7, third paragraph) along with one figure and a molecular structure from the SI. In general, this requirement is contrary to the requirement of Referee 2 to strengthen the evidence by making additional calculations. We disagreed with him and tried to justify it.

            I consider that the similarity of the distances of 2.32 and 2.333 Å is fortuitous and does not indicate that water interacts in similar fashion with C3H5+ and Na(OH2)5+ (whatever "similar" means). 

We have clarified what means "similar". They are not similar in fashion but are energetically similar because the frequencies of OH stretching vibrations of H2O molecules in III and in Na(OH2)6 are close, which reflect the strength of the OH2 bond with the cation. Changes made at the end of page 8 and at the end of the first paragraph on page 9 (marked in blue).

What is the evidence that water interacts with the C=C of the cation, rather than with the center of positive charge? 

If the distance between the O atom of H2O and the C2 and C3 atoms of the C3H7+ cation is the same, then the coordination of the O atom is directed geometrically to the middle of the C=C bond perpendicular to it. Since the "+" charge is concentrated on the C = C bond, it is obvious that the O atom interacts with it.  Nowhere is it said that the O atom interacts with the p-orbital of the C=C bond, since it is obvious that this is impossible.

That conclusion seems to come from Fig. 5, but why are there two oxygens in 5a, why is a carbon missing in 5b, and how do the neighboring {Cl11-} anions in the crystal (not shown) affect the structure?

Figure 5a show two O atoms because there are two overlapped locations of С3Н7+.H2O adduct (with a 50% probability of localization). Figure 5b show С3Н7+.H2O in one location. We did not understand where in Fig. 5b is a C atom lost? All C atoms are in place. Each C3H7+.H2O cation has the two shortest distances with the Cl atoms of the two {Cl11-} anions, being 2.854-2.993 Å (Figure),

which corresponds to the usual van der Waals interaction (the sum of the van der Waals radii of H and Cl atoms is 2.86 Å). All other distances between cation and anions (Н…Cl, C…Cl, O…Cl) exceed the corresponding van der Waals distances. That is, the influence of environmental anions on С3Н7+.H2O is ionically uniform. Anions are not shown in Figure 7 so as not to clutter the figure. Interactions of C3H5+ and C4H7+ cations with {Cl11-} anions are shown and discussed in our articles [15-17].

            The claim in the Abstract for a (meaningless) "short, strong, low-barrier (SSLB) asymmetric H–bond" between the C-H and an alcohol is not supported anywhere in the document and should be removed.  Is the self-citation (Ref. 18) really the shortest O-O distance in a hydrogen-bonded species?

The SSLB from abstract was removed. Our papers [19] and [20] (new list) collected all the literature known to us on proton disolvates L-H+-L from which it follows that the O…O distance in them varies in the range from 2.39 to 2.47 Å for L of different nature from strongly basic to weakly basic.

            The final conclusion is contradictory: "The formation of adducts ... by vinyl cations is unexpected, because according to quantum chemical calculations, they are energetically unfavorable and should not exist.  Thus, the alleged high reactivity of vinyl carbocations is an overestimation."  If quantum-chemical calculations imply that adducts of vinyl cations should not exist, whereas they do form, then the calculations are faulty, and vinyl cations are indeed reactive, as expected.

The sentence ”Thus, the alleged high reactivity of vinyl carbocations is an overestimation” was changed to “The very existence of these adducts means that the alleged high reactivity of vinyl carbocations is an overestimation.”

 Minor:

  • I recommend calling alcohol L an enol, to emphasize throughout the unusual nature of this alcohol.

 If everywhere in the text the word alcohol is replaced by enol, then this should be done in the title. It's hard for me to decide to do this.

(2) Do not call adduct III adduct A in the Abstract. 

Done.

(3) Why is adduct III discussed after I and II even though it is the first formed?

Because Results considers cations in the order in which crystals of their salts are separated from solutions with a decrease in their content of very small amounts of water. The cations are numbered in the order of their experimental formation from 1 to 3, and this order of consideration seems to be right.

In the Discussion, it is correct to consider the interaction of carbocations with an increase in the amount of water, and then the cations are considered in the order from 3 to 1. It is illogical and inconvenient to consider cations in the order from 1 to 3 both in the Results and in the Discussion.

Round 2

Reviewer 1 Report

After the revisions requested by me and other reviewers have been done, the manuscript is publishable in its current form.

Regarding my previous comment:

- Figure S1 can be moved to the main body of the paper from SI.

We did it.

-> I could not find that it had been done, but it was only a suggestion. This image is mentioned a couple of times in the text, and it would be easier for a reader if this image is available directly in the paper.

Figure 2, p. 4, should be changed to Figure 4.

Reviewer 4 Report

The authors have conscientiously revised their manuscript.  I have no further complaints.